# Chromosome-level assembly of the horseshoe crab genome provides insights into its genome evolution

Prashant Shingate [1], Vydianathan Ravi [1], Aravind Prasad[1], Boon-Hui Tay[1], Kritika M. Garg [2],
Balaji Chattopadhyay [2], Laura-Marie Yap[3], Frank E. Rheindt [2] & Byrappa Venkatesh [1✉]

The evolutionary history of horseshoe crabs, spanning approximately 500 million years, is characterized by remarkable morphological stasis and a low species diversity with only four extant species. Here we report a chromosome-level genome assembly for the mangrove horseshoe crab (*Carcinoscorpius rotundicauda*) using PacBio reads and Hi-C data. The assembly spans 1.67 Gb with contig N50 of 7.8 Mb and 98% of the genome assigned to 16 chromosomes. The genome contains five Hox clusters with 34 Hox genes, the highest number reported in any invertebrate. Detailed analysis of the genome provides evidence that suggests three rounds of whole-genome duplication (WGD), raising questions about the relationship between WGD and species radiation. Several gene families, particularly those involved in innate immunity, have undergone extensive tandem duplication. These expanded gene families may be important components of the innate immune system of horseshoe crabs, whose amebocyte lysate is a sensitive agent for detecting endotoxin contamination.

[1] Comparative and Medical Genomics Laboratory, Institute of Molecular and Cell Biology, A*STAR, Biopolis, Singapore 138673, Singapore. [2] Department of Biological Sciences, National University of Singapore, Singapore 117543, Singapore. [3] School of Applied Sciences, Republic Polytechnic, Singapore 738964, Singapore. ✉email: mcbbv@imcb.a-star.edu.sg

Horseshoe crabs (HSCs) are marine chelicerates belonging to the order Xiphosura and subphylum Chelicerata (phylum Arthropoda). There are only four extant species of HSCs, all belonging to the family Limulidae. Three of these (mangrove HSC, *Carcinoscorpius rotundicauda*; coastal HSC, *Tachypleus gigas*; and tri-spine HSC, *Tachypleus tridentatus*) inhabit tropical and subtropical Asia and one (Atlantic HSC, *Limulus polyphemus*) is found along the Atlantic coast of North America. HSCs are often cited as classical examples of 'living fossils'. They have existed for ~480 million years, with the oldest fossil recorded from the Upper Ordovician[1], yet their morphology has changed little. Recent fossils dating to the Middle Triassic, in which even soft tissue is exceptionally well preserved[2], point to an internal anatomy that has remained virtually unchanged over hundreds of millions of years. In addition to HSCs' remarkable morphological stability, their species diversity has remained low throughout their evolutionary history[3]. In fact, fewer than 50 fossil species are known even from the Carboniferous when the diversity of HSCs was at its peak[4].

Interestingly, analysis of HSC genomes has indicated the presence of a large number of duplicate genes with some families containing more than two paralogues which has led to the suggestion that the HSC lineage experienced two whole-genome duplication (WGD) events[5,6]. However, these genome assemblies were highly fragmented (contig N50 length 0.4–1.4 kb and scaffold N50 length 2.9 kb) and lacked synteny information that is essential for reliably inferring WGD events. Gene clusters, such as Hox clusters, are considered good indicators of the extent of WGDs in a lineage. Invertebrates typically contain a single Hox cluster with up to 10 members whereas genomes of spiders and scorpions, a major group of chelicerates collectively known as Arachnopulmonata, contain two Hox clusters owing to one round of WGD in their common ancestor after it diverged from the HSC lineage[7]. Due to the fragmented nature of the HSC genome assemblies by Nossa et al.[6] and Kenny et al.[5], no gene clusters could be recovered in their assemblies. However, recently a chromosome-level assembly (contig N50 length 1.7 Mb) of the tri-spine HSC was generated using long Nanopore reads[8]. This highly-contiguous assembly was found to contain two Hox gene clusters, raising the possibility that the HSC lineage has experienced only one WGD event akin to arachnopulmonates. Thus, the exact number of WGD events and the source of the large number of duplicate genes in HSCs remain uncertain.

The mud-dwelling HSCs have been successfully coexisting with myriads of microbial pathogens presumably due to the evolution of an efficient innate immune system. Their amebocytes are extremely sensitive to the lipopolysaccharides found in bacterial endotoxin which is why HSC blood lysate has been extensively used for detection of bacterial endotoxin contamination in injectable drugs and medical equipment[9]. However, in recent years many populations of HSCs have been under threat due to over-exploitation for bleeding[10]. In addition, extensive harvesting for human consumption (particularly in Asia) and habitat loss due to land reclamation and coastal modification have also contributed to their population decline[11]. These developments have resulted in the tri-spine HSC and the Atlantic HSC being, respectively listed as "Endangered" and "Vulnerable" in the IUCN Red List of threatened species (https://www.iucnredlist.org/) with the other two species being listed as "Data Deficient". Thus, there is a need to formulate strategies for managing and conserving existing HSC populations, yet very little is known about their population history and population genetic structure.

In the present study, we generate a high-quality, chromosome-scale genome assembly for the mangrove HSC using long PacBio reads and chromatin conformation capture data and reconstruct the population history of the species. This is the most contiguous assembly generated so far for any chelicerate. Analysis of this highly-contiguous genome provides evidence that suggests three rounds of WGD in the HSC lineage.

## Results

**Genome sequencing and assembly**. The genome size of the mangrove HSC was estimated to be 1.9 Gb based on the k-mer method (Supplementary Fig. 1, Supplementary Table 1) and the heterozygosity level was found to be 0.8%. Using single-molecule real-time PacBio reads (225 Gb, 118×), we first generated a contig-level assembly and then used Hi-C reads to organize them into a scaffold-level genome assembly. This was followed by extensive manual curation using Juicebox[12] and the Hi-C map whereby redundant contigs and mis-joins were removed to generate a high-quality genome assembly (Fig. 1a). The final genome assembly spans 1.67 Gb comprising 728 scaffolds with contig N50 and scaffold N50 lengths of 7.8 Mb and 102.3 Mb, respectively (detailed statistics given in Supplementary Table 2). Approximately 98% of the genome is present on the largest 16 scaffolds (Fig. 1a, Supplementary Table 3) which most likely correspond to 16 chromosomes of the mangrove HSC[13]. These scaffolds are hereafter referred to as chromosomes. To our knowledge, this is the most contiguous assembly among chelicerate genomes published to date (Supplementary Table 4). Searches for Benchmarking Universal Single-Copy Orthologs (BUSCO) revealed that the assembly contained complete sequences for 94.8% and partial sequences for 0.6% of the genes, with 4.6% of the genes missing. In addition, 16.7% of the genes were identified as duplicates. Approximately 93% of the clustered RNA-seq transcripts (626,967, size >500 bp) could be aligned to the assembly (>90% coverage and >90% identity), indicating that the assembly contained most of the gene sequences.

Transposable elements (TEs) comprise ~31% of the mangrove HSC genome (Supplementary Table 5). Using the MAKER annotation pipeline, we predicted 25,985 protein-coding genes (AED score of ≤0.5) that showed similarity to protein sequences in the NCBI-NR database (E-value cut-off of <1e−7). The average lengths of exons and introns of these genes are 237 bp and 4.3 kb, respectively. Protein domains were identified in 21,785 (~84%) of the predicted proteins. The top 20 Pfam domain families are given in Supplementary Table 6. The main features of the genome are summarised as a Circos plot in Fig. 1b. In general, an inverse relationship was seen between TE and coding sequence contents across all chromosomes (Fig. 1b).

**Population history of the mangrove HSC**. We estimated the neutral mutation rate of the mangrove HSC genome to be $1.37 \times 10^{-9}$ substitutions/site/year from a neutral tree based on four-fold degenerate (4D) sites (Supplementary Fig. 2). This estimate was used in pairwise sequentially Markovian coalescent (PSMC) analysis for determining the population history of the mangrove HSC. PSMC analysis revealed a history of population fluctuations for over ten million years and an entry into the Holocene with the historically lowest population size and genetic diversity (Fig. 2). A drastic decline in effective population size was apparent ~60,000 years ago coinciding well with the inception of the last ice age, during which a drastic reduction in the sea level would have led to considerable loss of shallow coastal habitat for this mud-dwelling coastal animal.

**Gene clusters in the genome**. Genes that are organized into clusters are often used to make inferences about WGD events in a particular lineage. One such cluster is the Hox gene cluster. Hox genes encode homeodomain-containing transcription factors that play a crucial role in defining identities of body segments and

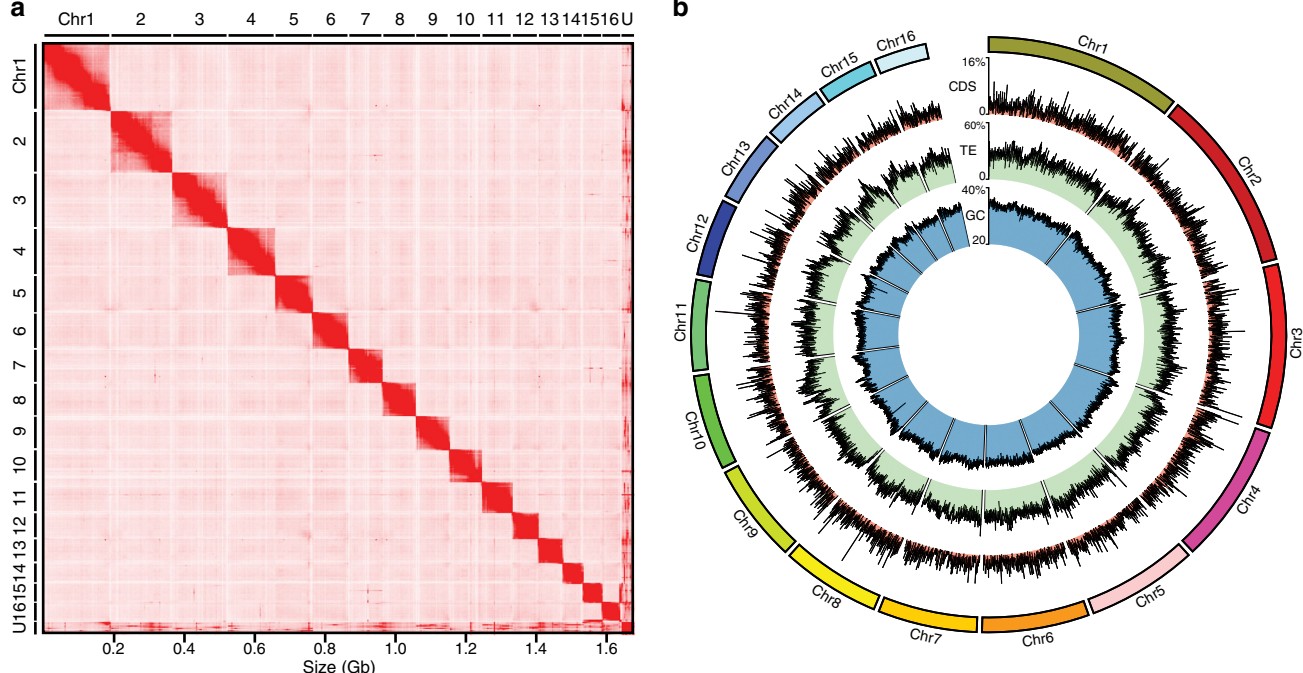

**Fig. 1 High-quality assembly of the mangrove HSC genome. a** Hi-C contact map of the mangrove HSC genome assembly. U, unplaced scaffolds. **b** CIRCOS plot showing the distribution of GC content, transposable elements (TE), and coding sequences (CDS) in the genome.

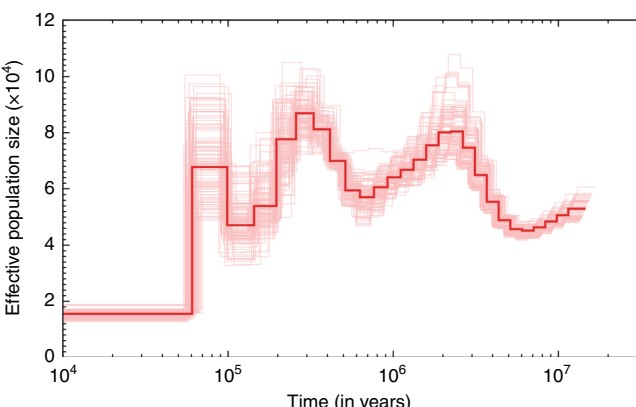

**Fig. 2 Population history of the mangrove HSC.** Pairwise sequentially Markovian coalescent (PSMC) model plot showing fluctuation in effective population size. The X-axis represents time (in years) before present on a logarithmic scale and Y-axis represents the effective population size. The bold red line represents the effective population size estimate obtained from the complete data. Uncertainty across this estimate obtained through bootstraps is represented by the scatter of thin red lines.

hence are considered attractive candidates for understanding the genetic basis of morphological diversity in animals[14,15]. Invertebrates typically contain a single Hox cluster with 10 genes. However, spiders and scorpions (Arachnopulmonata), a major group within chelicerates, possess two Hox clusters (see Fig. 3a) due to one round of WGD in their common ancestor[7]. In contrast, vertebrates such as mammals and other tetrapods contain four Hox clusters arising from two WGD events at the base of vertebrates[16,17]. Teleost fishes contain seven or eight Hox clusters as a result of an additional WGD event in the teleost ancestor[18]. Thus, the number of Hox clusters is a useful indicator for the number of WGD events in a lineage. We analyzed the mangrove HSC genome assembly and identified five Hox clusters containing

34 Hox genes in total (Fig. 3a). Two of the clusters (Hox-A and Hox-B) contain all 10 Hox genes found in arthropods whereas the other three are degenerate, containing between eight (Hox-C) and two Hox genes (Hox-E) (Fig. 3a). The five Hox clusters show considerable variation in size ranging from 1.3 Mb (Hox-A) to 122 kb (Hox-E) (Supplementary Fig. 3). This is the largest number of Hox genes and Hox clusters identified for any invertebrate. Their role in the development and phenotypic evolution of HSCs remains to be investigated. Interestingly, two of the clusters (Hox-A and Hox-C) are located on the same chromosome (chr_14), ~44 Mb apart, possibly because of a chromosomal fusion event that occurred after the WGD. Arthropod Hox clusters harbor four micro-RNAs, *mir-993*, *mir-10*, *mir-iab-4*, and *mir-iab-8*, with the last two present in the same region but on opposite strands. These genes were also duplicated along with the Hox genes, with *mir-10* being present in all five Hox clusters whereas *mir-993* and *mir-iab-4/8* were present in four and three Hox clusters, respectively (Fig. 3a).

In order to better understand the duplication history of Hox genes, we performed phylogenetic analysis using Hox genes from mangrove HSC and known Hox genes from Atlantic HSC, house spider and bark scorpion. The analysis indicated that all the duplication events in the HSC lineage occurred independent of the WGD event in spiders and scorpions (Arachnopulmonata) (Fig. 4), indicating that the WGDs in the HSC lineage occurred after it diverged from the ancestor of Arachnopulmonata. The presence of five Hox clusters in the mangrove HSC suggests that the HSC lineage experienced three WGD events followed by secondary loss of three Hox clusters, or alternatively, two WGD events followed by a segmental or chromosomal duplication.

MicroRNAs (miRNAs) are small noncoding RNAs that regulate gene expression through repression of mRNA translation. The *mir-71/mir-2* gene cluster is a particularly interesting invertebrate-specific miRNA cluster which has expanded in arthropods due to duplications. Spiders and scorpions were found to possess two of these gene clusters whereas the Atlantic HSC was found to contain seven loci for *mir-71* and/or *mir-2*[19].

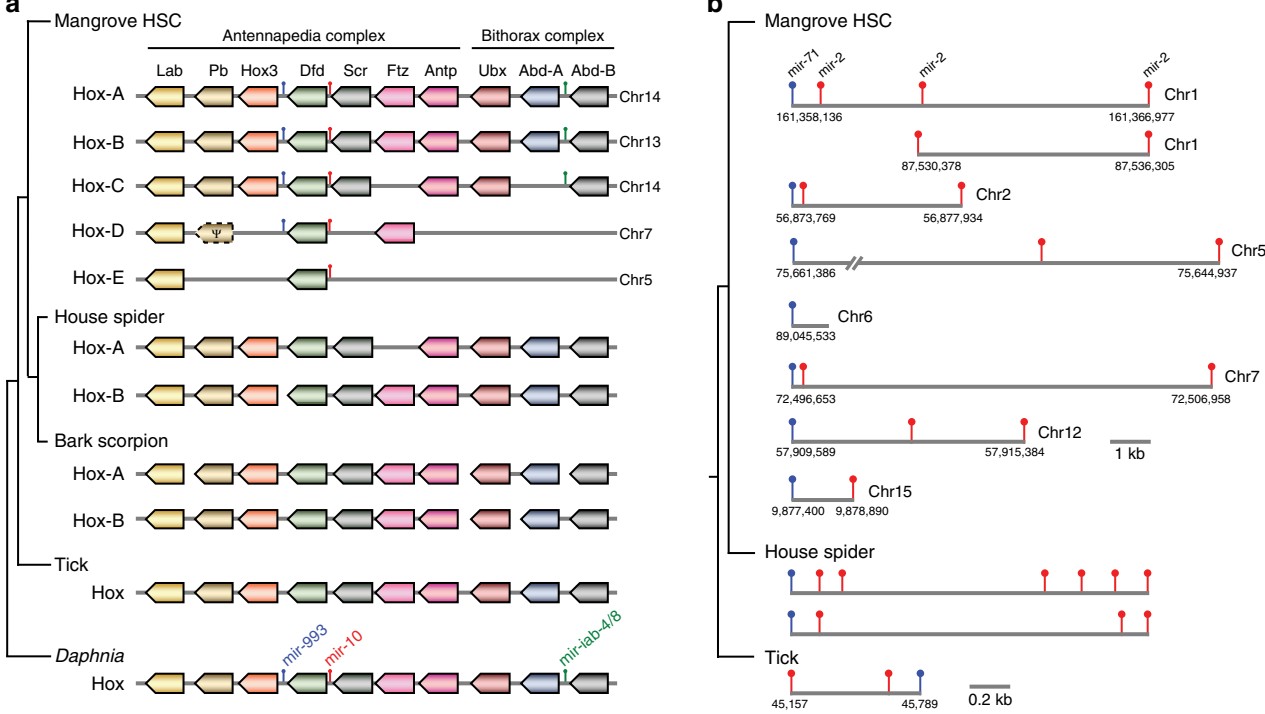

**Fig. 3 Gene clusters in the mangrove HSC genome. a** Hox gene clusters in the mangrove HSC compared to those in house spider, bark scorpion, tick, and *Daphnia*. The positions of four miRNAs (*mir-993, mir-10, mir-iab-4*, and *mir-iab-8*) are also shown, the latter two being located at the same position but on opposite strands. **b** miRNA *mir-71* and *mir-2* gene loci in the mangrove HSC genome. House spider and tick miRNA gene loci are shown for comparison. The house spider miRNA loci are not drawn to scale.

We searched the mangrove HSC genome for these miRNAs and identified potentially eight loci on seven chromosomes (Fig. 3b). However, it is possible that the two *mir-2* genes on chromosome 1 (at 87.53 Mb) could be part of the *mir-71* and *mir-2* locus on chromosome 15 in which case there would be only seven paralogous loci.

We also searched for other gene clusters present as six or more copies in the genome and found several such instances. We present ten such instances of clusters: six with seven paralogous loci (Fig. 5 and Supplementary Fig. 4–9) and four with six paralogous loci (Supplementary Fig. 10–13). One of these, the "*Mical3L* locus" (Fig. 5c and Supplementary Fig. 7), contains potentially eight paralogous loci. These multiple copies of paralogous loci containing clusters of mostly unrelated genes are particularly informative in shedding light on the history of gene/genome duplication in the HSC lineage. For example in the "*Pde4L* locus", two genes (*Pde4L* and *Bmp5L*) (Fig. 5a and Supplementary Fig. 4) are present in seven copies while the rest are present in only two to four copies, which is likely due to the differential loss of duplicate genes in the paralogous loci after duplication event. Likewise, in the "*Notch*L locus", *Notch*L is found in seven copies (Fig. 5b and Supplementary Fig. 5) while the rest are present in only two to five copies due to the differential loss of duplicated genes in paralogous loci. The presence of these paralogous loci with up to eight copies located mostly on different chromosomes in the mangrove HSC genome is more consistent with three rounds of WGD followed by differential secondary loss of duplicated genes than segmental duplications or tandem gene duplication followed by transloca-tion. It is interesting to note that five of the seven "*Fas1L* loci" are linked to the five Hox clusters (Fig. 5d). This extends the syntenic block of the Hox clusters described above and suggests that the ancestral chromosomal segment containing the "*Fas1L* locus" and

the Hox cluster underwent duplication together followed by loss of two Hox clusters linked to the Fas1L loci on chromosomes 2 and 4.

**Comparison with the tick genome**. To gain further insights into the WGD events in the mangrove HSC, we compared its genome with a chelicerate that has not experienced WGD, the black-legged tick (*Ixodes scapularis*). The divergence of the HSC lineage from the tick lineage is a rather ancient event that occurred during the early Cambrian, ~530 Ma[20]. Of the two tick genome assemblies available[21,22], the one by Miller et al.[22] (https://www.ncbi.nlm.nih.gov/assembly/GCA_002892825.2) has a much better contiguity (contig-level assembly with contig N50 of 835 kb) and hence we chose this assembly. However, this tick assembly is much less contiguous than the mangrove HSC assembly (scaffold N50: 102.3 Mb). We identified orthologues between the two genomes using InParanoid and found that out of 24,054 genes in the tick genome 3825 genes had 1-to-1 orthologues in the man-grove HSC genome whereas 1461 possessed 2–8 orthologues (972, 1-to-2; 319, 1-to-3; 115, 1-to-4; 40, 1-to-5; 10, 1-to-6; 5, 1-to-7; and 0, 1-to-8). Despite this limited number of 2–8 ortho-logues (expected products of three rounds of WGD) between tick and mangrove HSC and the fragmented nature of the tick gen-ome assembly, we found several instances of a single tick syntenic block mapping to multiple chromosomes in the mangrove HSC. Three such examples are presented in Fig. 6a–c. This pattern of inter-digitated distribution of mangrove HSC orthologues on different chromosomes suggests that these syntenic blocks of genes in mangrove HSC may be the result of WGD rather than segmental or tandem duplication events.

**Paralogous segments in the mangrove HSC genome**. WGD events generate duplicated segments in the genome which can be

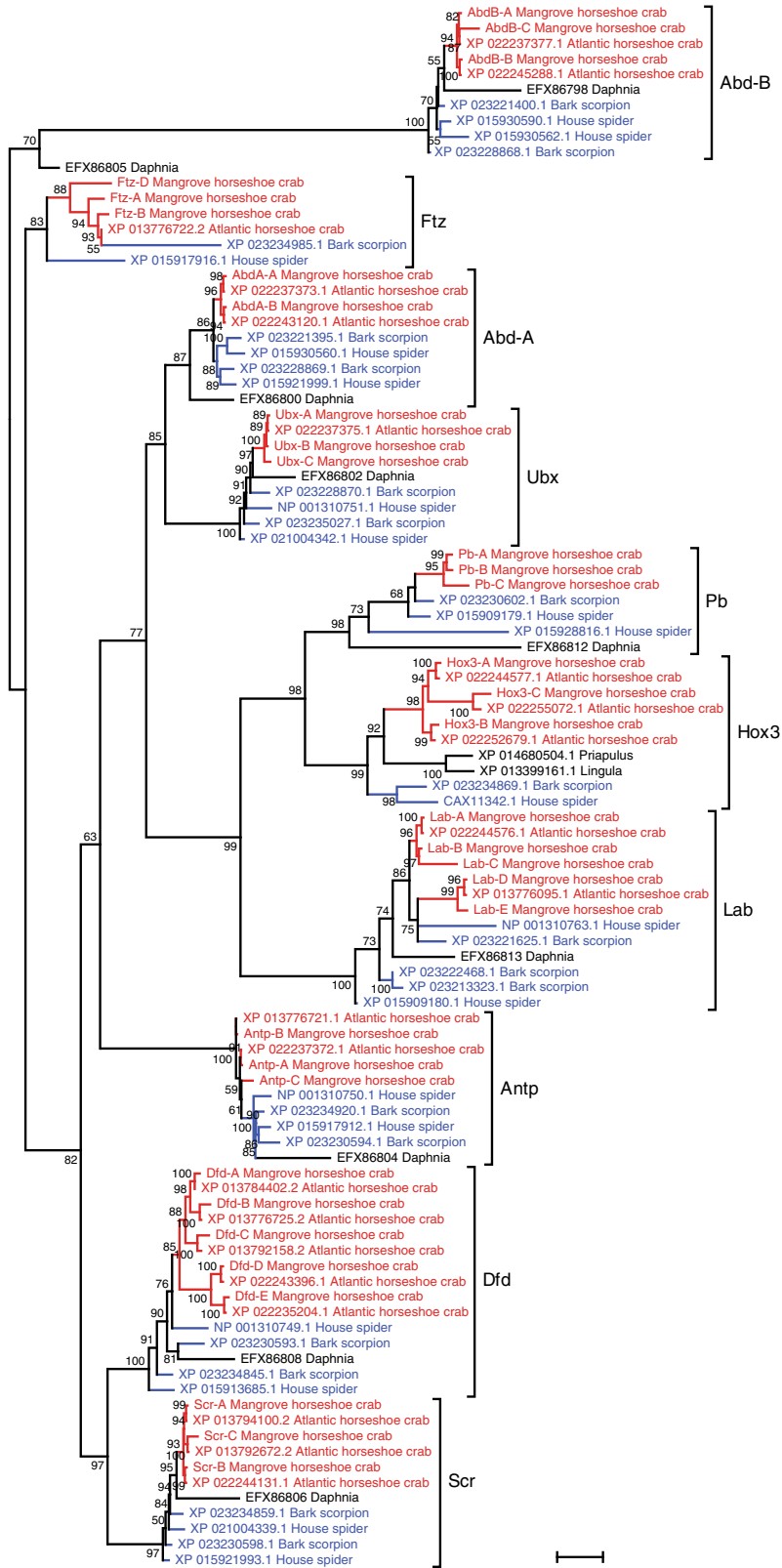

**Fig. 4 Phylogenetic tree of Hox genes from horseshoe crabs and other selected arthropods.** Maximum likelihood tree showing the phylogenetic relationship of Hox genes from horseshoe crabs (red font), spider and scorpion (blue font) and outgroup species (black font). Values at the nodes represent bootstrap support percentages. The scale bar denotes one substitution per site.

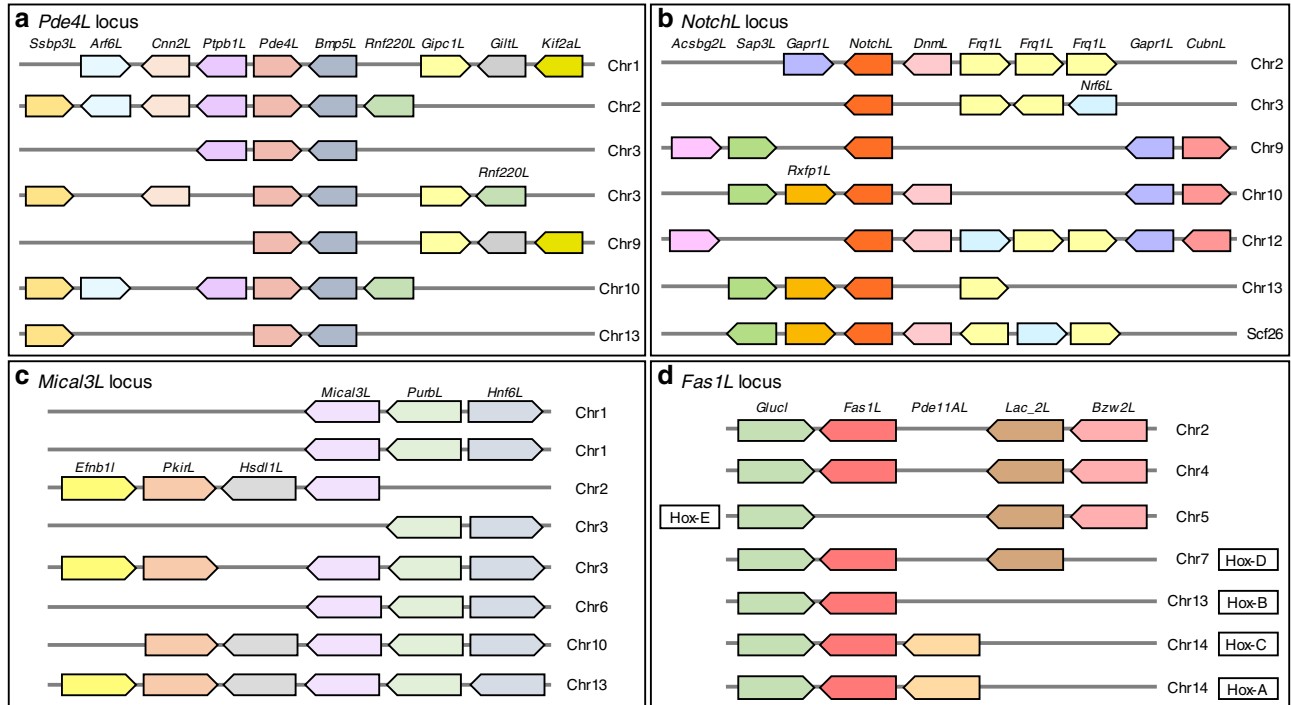

**Fig. 5 Syntenic blocks with multiple paralogous loci in the mangrove HSC genome. a** "Pde4L locus", **b** "NotchL locus", **c** "Mical3L locus" and **d** "Fas1L locus". Chromosomal coordinates of these loci and their corresponding homologs in bark scorpion and tick are shown in Supplementary Figs. 4–7. Five of the Fas1L loci (**d**) are linked to the Hox clusters as shown. While the Hox-A, Hox-B, and Hox-C clusters are located within 350 kb from their corresponding Fas1L locus, the Hox-D and Hox-E clusters are located ~10 Mb and 4 Mb away, respectively (see chromosomal coordinates in Supplementary Figs. 3 (Hox clusters) and 6 (Fas1L locus).

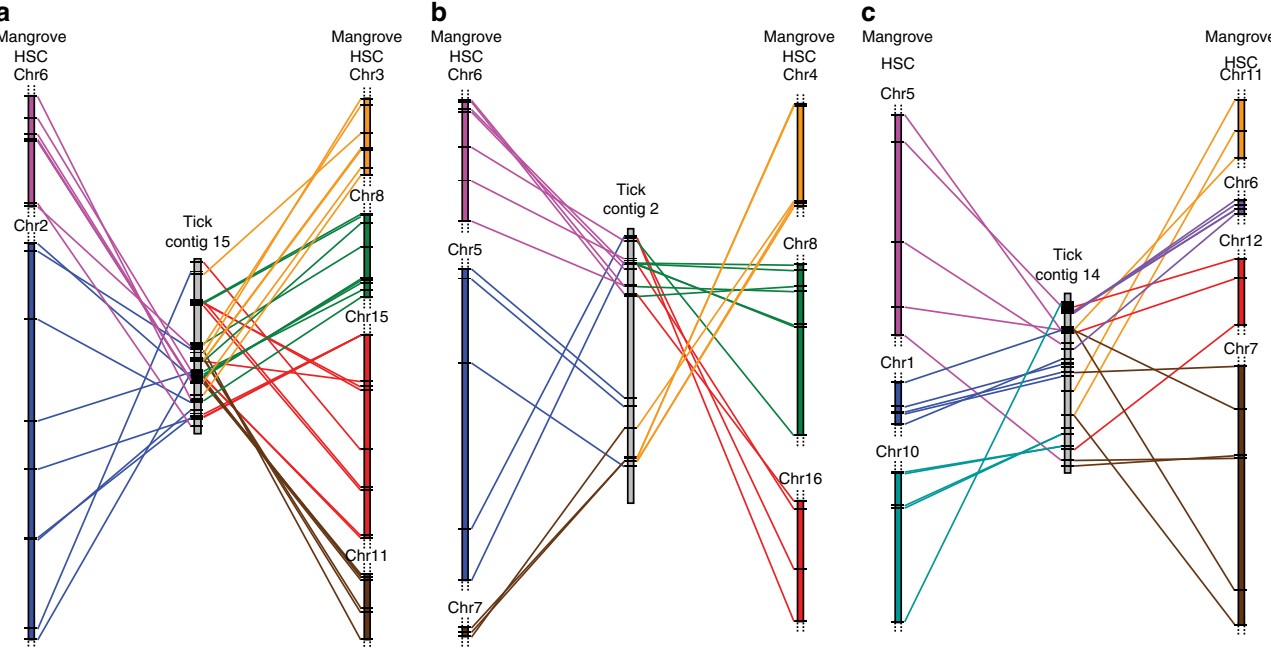

**Fig. 6 Syntenic relationship between tick contigs and mangrove HSC chromosomes.** Relative positions of orthologues between tick contig 15 (**a**; 5.5 Mb), contig 2 (**b**; 12.4 Mb) and contig 14 (**c**; 5.5 Mb), and mangrove HSC chromosomes are shown. Colored vertical bars represent contigs (tick) or chromosomes (mangrove HSC) whereas the horizontal black lines represent orthologous genes. Dotted lines indicate that the chromosomal region extends beyond what is shown.

identified as 'paralogons', i.e. syntenic blocks of paralogous genes shared between chromosomes. The presence of such paralogons throughout the genome suggests that the duplicate genes are the result of WGD rather than independent tandem duplications

followed by translocation in which case the paralogous genes would be randomly distributed in the genome. In case the genome had experienced independent duplication of one or a few chromosomes instead of a WGD, it would exhibit strong

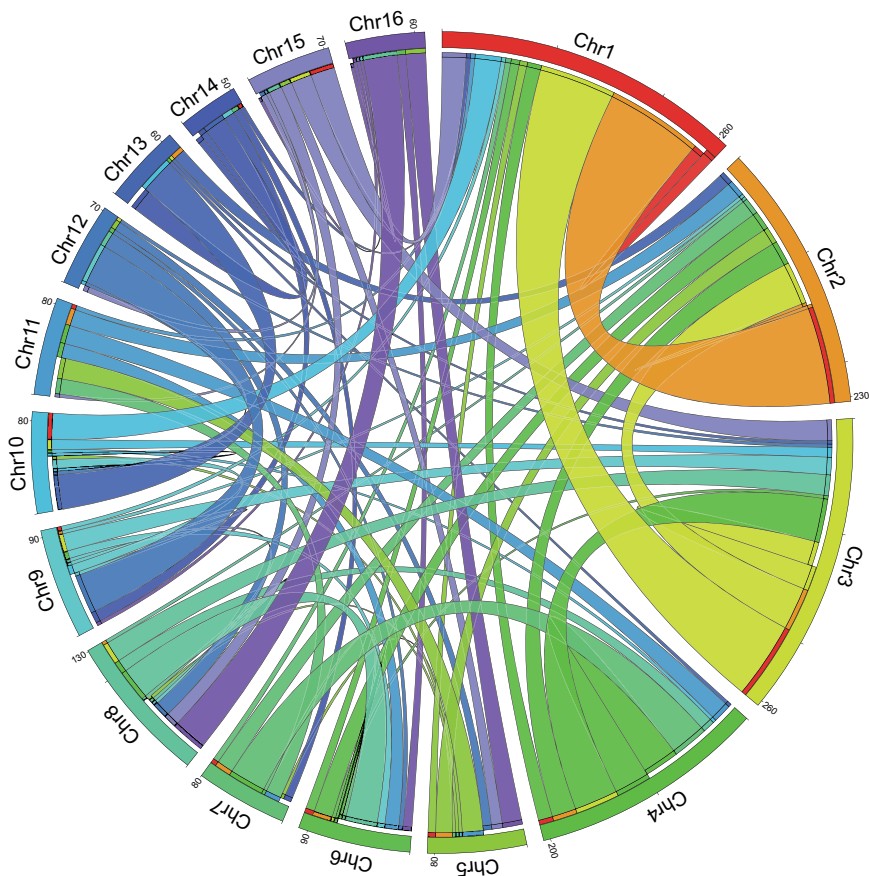

**Fig. 7 Paralogous segments in the mangrove HSC genome.** Circos plot showing paralogous regions (paralogons) in the mangrove HSC genome. Numbers on the chromosomes represent the number of anchor points, i.e. sets of syntenic paralogous genes. The thickness of the bands is proportional to the number of anchor points shared between two chromosomes. Note that the position of the bands on the chromosome do not indicate the chromosomal location of the anchor points.

paralogous relationship between just one or a few pairs of chromosomes. To verify these possibilities, we identified paralogous segments in the mangrove HSC genome and then visualized their distribution in the genome using a Circos plot. Tandem duplicates were first removed from the protein dataset retaining only a single representative for each locus. Paralogous families were then generated by BLAST of this protein dataset against itself followed by clustering. i-ADHoRe[23] was next used to identify syntenic blocks. Visualization of paralogons using the Circos plot (Fig. 7) showed that each mangrove HSC chromosome contains genes whose paralogues map to multiple other chromosomes. This pattern is more consistent with WGDs rather than independent chromosomal or tandem gene duplication events and supports the hypothesis that the HSC lineage has experienced three rounds of WGD.

In order to verify this hypothesis, we estimated the rate of transversions at four-fold degenerate sites (4DTv) for paralogous gene pairs in the mangrove HSC genome. 4DTv analysis is a conservative estimate of genetic divergence and is less susceptible to saturation effects compared to the synonymous substitution rate (Ks) and is preferred for inferring the number of ancient WGD events[24,25]. We performed 4DTv analysis of paralogous gene pairs in mangrove HSC followed by fitting of univariate normal mixture models to the corrected 4DTv distribution. Bayesian Information Criterion was then used to identify the best-fit model. The best-fit model showed four distinct population components in total, of which the first peak (mean value 0.02, not shown) represents recent small-scale duplication events (Fig. 8).

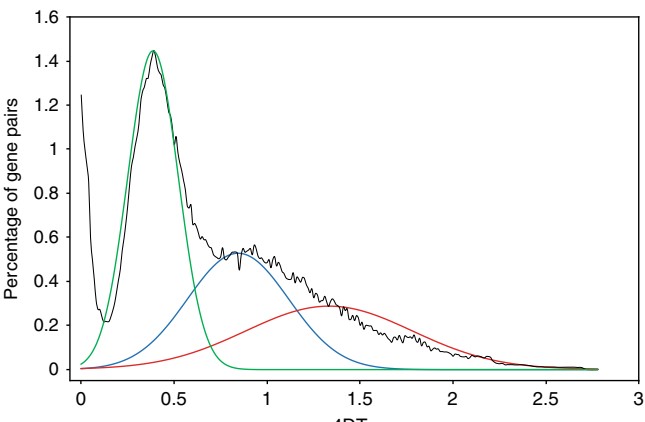

**Fig. 8 Whole-genome duplications in the mangrove HSC genome.** 4DTv analysis showing the relative age of paralogues in the mangrove HSC genome. Three peaks were identified using the Bayesian information criterion. The populations under red, blue, and green curves represents 4DTv values of paralogue pairs belonging to the first, second, and third rounds of WGD, respectively.

The remaining three peaks (green, blue and red; mean values of $0.39 \pm 0.14$, $0.85 \pm 0.27$ and $1.33 \pm 0.45$; Fig. 8) represent three WGD events and provide further support for the inference that the HSC lineage experienced three rounds of WGD.

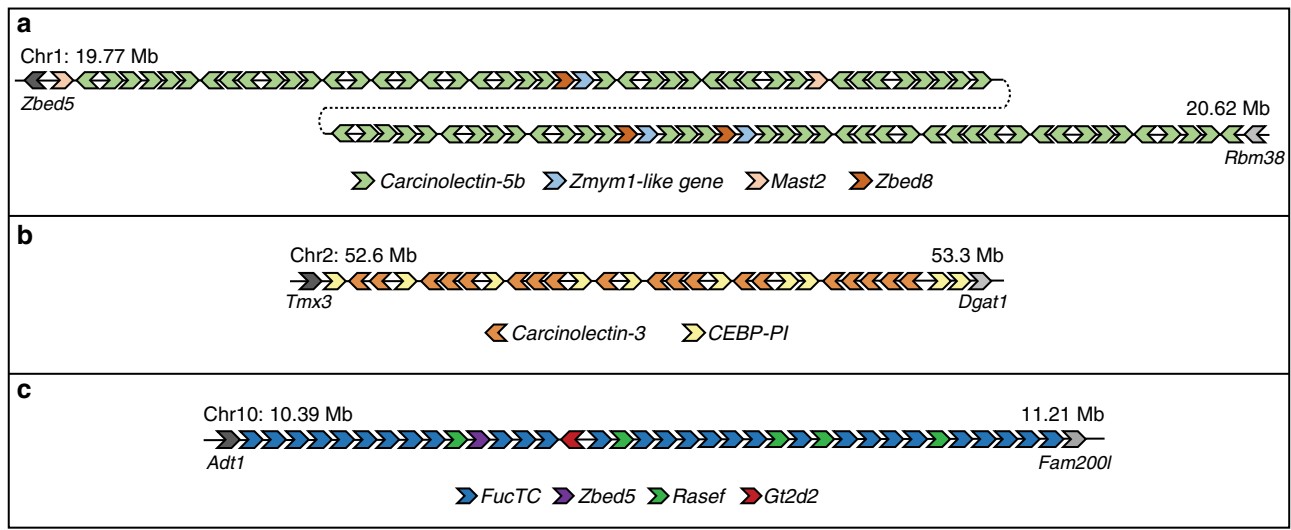

**Fig. 9 Innate immune system-related tandem gene clusters in the mangrove HSC genome. a** *Carcinolectin-5b* locus, **b** *Carcinolectin-3/CEBP-PI* locus and **c** *FucTC* locus. Carcinolectin is the mangrove HSC homolog of tachylectin that was previously identified in the tri-spine HSC. The figure is based on manual annotation of these loci which helped to identify more genes than those predicted by MAKER annotation.

**Tandem gene clusters**. Previous analyses of HSC genomes have indicated that they contain a large number of paralogous genes[5,6,26]. These duplicate genes could be the result of WGD, segmental duplications or small-scale tandem gene duplications. Lineage-specific tandem gene duplication events are a particularly interesting category as they are often related to the unique biology of a taxon. We performed a systematic search for tandem gene clusters in the mangrove HSC genome and identified 230 such clusters containing three or more genes (Supplementary Data 1). A search for these genes in the genomes of bark scorpion, house spider, and tick using TBlastN indicated that majority of these tandem gene clusters are unique to the mangrove HSC (Supplementary Data 1).

Gene ontology enrichment analysis of the tandem genes showed that GO terms such as monooxygenase, oxidoreductase and fucosyltransferase activity and glycosylation are enriched (FDR ≤ 0.05 and *P*-value ≤ 0.01; Supplementary Table 7). The list includes several genes associated with the innate immune system. The largest cluster found in the genome is that of the *carcinolectin-5b* gene which comprises 79 genes spread across ~850 kb (Fig. 9a). Carcinolectin-5b is a homolog of the plasma-derived lectin, tachylectin-5b, identified in the tri-spine HSC. Tachylectin-5 agglutinates Gram-positive and Gram-negative bacteria[27] while carcinolectin-5b has been shown to bind and stabilize interactions between galactose-binding protein (GBP), C-reactive Protein (CRP) and lipopolysaccharides (LPS) present in invading pathogens, leading to the activation of the classical complement pathway[28]. A related gene cluster is that of *carcinolectin-3* comprising 19 copies of the gene interspersed with 10 copies of the *Carcinoscorpius endotoxin-binding protein protease inhibitor* (*CEBP-PI*) gene, together spanning ~670 kb (Fig. 9b). The homolog of carcinolectin-3 in the tri-spine HSC, tachylectin-3, is located in the secretory granules of amebocytes, where it recognizes Gram-negative bacteria via unique structural units of O-antigens on LPS and initiates a coagulation cascade event[29]. Carcinolectin-3 is expected to perform a similar function in the mangrove HSC. The homolog of CEBP-PI in the Atlantic HSC, LEBP-PI, is located in the secretory granules of amebocytes along with other enzymes involved in the coagulation cascade. This protein possesses an anti-tryptic activity and a high binding affinity to endotoxins, in particular to LPS. It has been proposed

that binding of LEBP-PI to LPS suppresses exocytosis of granules from the amebocytes which results in termination of the clotting process after the pathogens are confined in the clot[30]. The mangrove HSC genome also contains a cluster of 29 *FucTC* genes (alpha1,3-fucosyltransferase C) spanning ~820 kb (Fig. 9c). Fucosylation is a major type of glycosylation process which involves transfer of a fucose unit from GDP-fucose to their substrates. In mice, FucTC has been shown to regulate leukocyte trafficking between blood and the lymphatic system through its involvement in selectin ligand biosynthesis[31].

Besides immune genes, some genes important in the maintenance and stability of the genome were also found to be expanded in the mangrove HSC. For instance, there are four tandem copies of a p53-like gene on chromosome 4 (120.8–121.1 Mb) in addition to a single p53-like gene each on chromosomes 8 and 12. The p53-like genes (p53/p63 and p73) are referred to as guardians of the genome as they play a crucial role in the fidelity of DNA replication and cell division by initiating cell cycle arrest, senescence or apoptosis[32]. The p53 protein is also known to confer genomic stability by preventing conflicts between transcription and replication processes[33]. The large-bodied elephant possesses 20 copies of the *p53* gene which confer elephant cells an enhanced TP53-dependent DNA-damage response[34]. Besides the p53-like gene cluster in the mangrove HSC, a cluster of three genes encoding the mini-chromosome maintenance complex-binding protein (MCMBP) is present on chromosome 15 (47.21–47.23 Mb). MCMBP facilitates cohesion of sister chromatids which is essential for post-replicative homologous recombination repair and is thus important for chromosomal stability[35,36]. The expansion of these gene families related to genome stability and maintenance might have contributed to the high-fidelity of DNA replication and stable chromosomes.

One of the enriched GO terms associated with tandem gene clusters is related to ecdysis (Supplementary Table 7). The mangrove HSC contains a cluster of four eclosion hormone (EH) encoding genes on chromosome 1 (143.96–143.99 Mb), besides a single gene present on chromosome 3. EH is a key regulator of ecdysis—a process consisting of periodic shedding of the cuticular exoskeleton that occurs during growth in Ecdysozoa[37]. HSCs, with a prosomal width of 12–28 cm[38], are the largest among the

extant chelicerates. Expansion of the *EH* gene family may be related to an efficient ecdysis system required by these large-bodied chelicerates.

## Discussion

HSCs are ancient marine chelicerates that have changed very little over 500 million years of evolution. Despite their long evolutionary history, HSCs are currently represented by only four extant species. These species shared a common ancestor ~135 million years ago[39]. Whole genomes of three of them (Atlantic, mangrove, and tri-spine HSCs) have been previously sequenced using either short (Illumina and Roche 454) or long (Nanopore) reads[5,6,8,26,40]. Analysis of these genome assemblies suggested that they experienced one or two rounds of WGD, with one study proposing that one of the WGD events in the tri-spine HSC occurred after it diverged from the Atlantic HSC lineage[40]. In the present study, we have generated a high-quality chromosome-level genome assembly for the mangrove HSC using PacBio long reads and chromatin conformation capture (Hi-C) data followed by extensive manual curation. This chromosome-level assembly with a contig N50 of 7.5 Mb is the most contiguous assembly generated for a HSC so far (Supplementary Table 4). Analysis of this highly contiguous assembly has uncovered several lines of evidence that argue more in favor of three rounds of WGD in the HSC lineage than independent tandem duplications followed by translocation or large-scale segmental duplications. Phylogenetic analysis of the duplicated *Hox* genes indicated that the duplication events occurred prior to the divergence of the mangrove HSC and Atlantic HSC, implying that the duplications occurred in the common ancestor of the four extant species of HSCs. Thus, the timing of the WGD events can be placed between 135 Ma and 500 Ma (i.e. the approximate divergence time of the HSC lineage from its sister chelicerate lineage). In addition to the WGD events, analysis of the mangrove HSC assembly also highlighted several tandem gene clusters that are specific to the HSC lineage and likely underlie some of the unique biological traits of HSCs, such as a highly efficient innate immune system. The intact recovery of long stretches of tandem gene copies in our assembly is likely due to the long PacBio reads used, which could be assembled into long gap-free contiguous sequences (longest contig is 50.2 Mb).

Pairwise sequentially Markovian coalescence (PSMC) analysis of our mangrove HSC genome provided unique insights into fluctuations in population size and genetic diversity of this species over the last few million years. These analyses are particularly accurate for timescales involving the Quaternary (i.e., 2.5 Ma onwards) and point to a history of population fluctuations in the HSC throughout the epoch of glacial cycles that have dominated our planet during the Pleistocene (Fig. 2). Most importantly, we detected an especially pronounced decline in effective population size around ~60,000 years ago (Fig. 2), coinciding with the onset of the most recent ice age, from which this species has since not recovered. The mangrove HSC critically depends on shallow coastal waters for its survival. In its Southeast Asian home range, the extent of shallow coastal habitat is greatly diminished every time the planet undergoes a glacial period, when global sea levels drop by up to ~120m, exposing the shallow waters of the Sunda Shelf as land. During glacial periods, such as between ~60,000 and 20,000 years ago, most of the mangrove HSC's present habitats would have been covered by land, while the sea coasts at that time would have been greatly reduced in extent and in suitability owing to steep drops in sea depth at the margin of the Sunda Shelf[41]. The lack of recovery in genetic diversity during the present interglacial (i.e., post-20,000 years ago) may be due to lag times associated with the slow evolutionary rate of this 'living

fossil'. Interglacials with higher sea levels, such as the present time, are crucial periods for HSCs to replenish their genetic diversity. Yet the mangrove HSC's present effective population size is as low as it has been in ~10 million years, while human activity only leads to their further decline. Habitat conservation and harvesting bans are urgently needed to ensure that the present human-dominated interglacial does not spell the end of an organism that has otherwise survived for almost half a billion years.

WGD events result in doubling of the genome, thereby providing additional, initially redundant genetic material that can be potentially targeted for evolutionary innovations. WGDs are found at the base of notable radiations, such as vertebrates (>60,000 species), teleosts (~30,000 species) and flowering plants (>350,000 species), and are therefore thought to be causally related to adaptive radiation and morphological diversification[42]. Indeed the common ancestor of spiders and scorpions, a major group of chelicerates, also experienced a WGD event ~450 Ma independent of the HSC lineage[7], and both spiders and scorpions are known to be species-rich (~40,000 extant species together) and exhibit extensive morphological diversity[43]. The evolutionary history of HSCs in a way parallels the evolutionary history of teleost fishes which have experienced a teleost-specific WGD on top of two WGD events that occurred at the origin of vertebrates. Yet, HSCs have been characterized by very low levels of species diversity throughout their evolutionary history[3] culminating in only four extant species. In addition, HSCs have also exhibited a high degree of morphological stasis during their long evolutionary history. Thus, it seems that although WGDs can generate genome-wide redundant genetic material upon which evolution may act, a WGD event by itself is not sufficient in driving species diversity. Adaptive radiation and morphological diversification are complex processes involving interactions between multiple environmental and genetic factors. For example, the adaptive radiation of cichlid species in East African lakes has been attributed to pre-existing genetic diversity in the lineage and the diverse ecological opportunities offered by the newly formed lakes[44]. It appears that HSCs, which have few natural predators and inhabit nutrient-rich coastal marine habitats, did not venture out into any other environment. Thus, there was probably no need for them to diversify or adapt to new environments.

Analysis of the mangrove HSC genome identified many tandem gene clusters that are specific to the HSC lineage. Some of these expanded gene families, such as carcinolectin-5, carcinolectin-3, CEBP-PI, and FucTC, play important roles in innate immunity. Mangrove HSCs live in muddy environments and are constantly exposed to a variety of pathogens. In order to survive and propagate successfully in such an environment they have evolved a sophisticated innate immune system response network for recognition, immobilization and killing of pathogens[45]. In fact, it is known that their amebocytes are extremely sensitive to bacterial lipopolysaccharides and hence the HSC amebocyte lysate is routinely used as a sensitive agent for detecting bacterial endotoxin contamination[9]. The expanded innate immune system gene families identified in our study are likely to be important components of the HSC immune response network and probably help them mount a rapid and effective immune response to counteract pathogens.

## Methods

**Genome sequencing.** An adult mangrove HSC specimen was obtained from the aquaculture facility of Republic Polytechnic, Singapore. Four tissue samples (egg, gill, leg, and muscle) were collected and flash-frozen in liquid nitrogen. High-molecular-weight genomic DNA extracted from the muscle tissue was used to prepare long insert genomic libraries followed by SMRTbell™ templates. The SMRTbell™ templates were sequenced using P6C4 sequencing chemistry on a PacBio Sequel resulting in the generation of ~225 Gb reads (N50 read length

~17 kb) which corresponds to approximately 118× coverage of the genome. In addition, PCR-free Illumina libraries with insert sizes varying from 350 to 470 bp were prepared using Kappa Hyper Prep kit (Kapa Biosystems, South Africa). An Illumina HiSeq 4000 platform was used to sequence these libraries to generate ~226 Gb of 150 bp paired-end reads which translates to ~119× coverage of the genome.

**Contig-level assembly.** The FALCON-Unzip Assembler[46] was used to assemble the SMRT reads with the help of DNAnexus, Inc. (San Francisco, CA, USA). This step produced two contig-level genome assemblies, the primary assembly and the accessory assembly. The contig-level primary assembly was polished using raw SMRT reads with Arrow version 2.2.1[47] followed by an additional round of error-correction using ~226 Gb of Illumina reads with Pilon version 1.21[48]. The primary assembly represents a near-complete haploid assembly and is not supposed to contain any redundant contigs. However, we encountered several duplicate (heterozygous) contigs in the assembly. Additionally, several contig pairs were found to overlap by 10 kb to several Mb at the terminal regions. An in-house script was used to filter out duplicate copies of contigs using the criteria: ≥95% identity and ≥80% coverage with respect to the smaller contig. Another in-house script was used to join >30 kb overlapping terminal regions of the contigs with ≥95% identity.

**Hi-C aided chromosome-level whole-genome assembly.** An Arima Hi-C library was prepared by Arima Genomics (San Diego, CA) using the mangrove HSC muscle tissue and a proximity ligation technology that captures long-range promoter-enhancer interactions. The library was sequenced on an Illumina sequencing platform to generate read pairs of 150 bp length. These Hi-C library reads were used for scaffolding the FALCON non-redundant primary assembly with the SALSA (https://github.com/marbl/SALSA). A single round of gap-filling was performed on these scaffolds using all error-corrected SMRT reads and PBJelly from PBSuite version 15.8.24 (https://sourceforge.net/projects/pb-jelly/). To address any errors in this near-chromosome-level genome assembly, we used Hi-C contact maps generated using Juicer v1.5.7 and 3D-DNA pipeline version 180922[49]. We visualized the Hi-C contact map and performed extensive manual curation using Juicebox version 1.11.08[12] (available at https://github.com/aidenlab/Juicebox/wiki/Download) to remove residual duplicate contigs and fix mis-joins. These corrections were incorporated into the assembly using the post-review option within 3D-DNA pipeline. The resulting assembly was subjected to three rounds of error correction using Illumina reads and Pilon program.

**RNA-seq.** Total RNA was extracted from egg, gill, leg, and muscle using the TRIzol reagent (Invitrogen, Carlsbad, USA), treated with DNase I (TaKaRa Bio Inc, Shiga, Japan) and purified using the RNeasy Mini Kit (QIAGEN, Hilden, Germany). An RNA-seq library was constructed using Ribo-Zero Gold reagent and ScriptSeq v2 Library Preparation Kit (Epicentre, Madison, USA). The library quality and quantity were analyzed on an Agilent 2100 Bioanalyzer. Sequencing was performed using 151 cycles on an Illumina NextSeq machine. A total of ~82–149 million paired-end reads from each tissue were assembled de novo using Trinity version 2.2.0[50].

The Trinity-assembled RNA-seq transcripts (4.43 million) from the four tissues were clustered using CD-HIT (http://weizhongli-lab.org/cd-hit/) with a sequence identity threshold of 97% and alignment coverage of 80% for the shorter sequence. The clustered sequences (2.3 million) were subjected to BLASTX searches (E-value cut-off: 1e−7) against chelicerate proteins from the NCBI NR database after removing partial and fragmented sequences (total 229,473 proteins). Transcripts with a BLASTX hit were translated using TransDecoder version 5.0.1 (available at https://github.com/TransDecoder/TransDecoder) and the protein sequences were again clustered using CD-HIT. The clustered proteins were searched against chelicerate proteins in the NCBI database using BLASTP and 12,137 unique 'full-length' protein sequences that showed ≥80% subject coverage were identified.

**Assembly quality and completeness.** The completeness of the PacBio-HiC assembly was assessed by using the 1066 Arthropod gene set from OrthoDB v9 and the Benchmarking Universal Single-Copy Orthologs (BUSCO) version 2.0[51]. The completeness was further assessed by aligning CD-HIT clustered TRINITY transcripts (626,967, size >500 bp) to the assembly using BLAT[52].

Recently, a chromosome-scale genome assembly was generated for the tri-spine HSC (*Tachypleus tridentatus*)[8]. We intended to align this genome to our assembly to assess the quality of the mangrove HSC genome. However, all-against-all BLASTP analysis of the predicted protein sequences of tri-spine HSC showed that of the 34,966 proteins predicted, 4855 had duplicate copies (≥98% identity and ≥70% coverage) in the assembly suggesting considerable amount of redundancy and mis-assembly of the genome. Also, as reported in the manuscript[8], the assembly contains only two Hox gene clusters as compared to five clusters identified by us in the mangrove HSC assembly suggesting that many genes are missing in the tri-spine HSC assembly. Therefore, we decided that comparison to tri-spine HSC genome will not be informative.

**Genome size and heterozygosity level.** Illumina trimmed reads were used as input to calculate the distribution of k-mer copy number (KCN). We tried a range of k-mers ranging from 15 to 57, and selected 31 which gave two distinct peaks of k-mer frequency distribution (Supplementary Fig. 1) to obtain the KCN distribution using Jellyfish version 2.2.6 (https://github.com/gmarcais/Jellyfish). The first peak (KCN = 46) represents the heterozygous single copy k-mer while the second peak (KCN = 92) represents the homozygous single copy k-mers in the genome (Supplementary Fig. 1). The genome size was estimated using the Lander-Waterman method[53] based on estimated read-depth (RD).

All k-mers with very small KCN values were considered as sequencing errors. Hence combined length of k-mers with a KCN value less than or equal to the first minima in the KCN distribution were subtracted from the estimated genome size. The KCN distribution obtained from Jellyfish at k-mer value 31 was used as input to predict the heterozygosity level. The heterozygosity level was calculated using GenomeScope (https://github.com/schatzlab/genomescope).

**Prediction of repetitive sequences.** RepeatModeler version 1.0.10 (http://www.repeatmasker.org) was used to generate a de novo repeat library from the mangrove HSC PacBio-HiC assembly. The resulting repeat library contained 366 repetitive elements. This repeat library was then combined with known Chelicerate repeats in the RepBase version 22.05 (https://www.girinst.org/server/RepBase/). This final mangrove HSC-specific repeat library was used to estimate the repeat content of the assembly.

**Genome annotation.** The repetitive regions in the genome were masked using RepeatMasker version 4.0. Evidence-based gene prediction was performed followed by ab initio gene prediction using the MAKER pipeline version 2.31.9[54]. Approximately 12,000 full-length proteins obtained from the mangrove HSC transcriptome was used for evidence-based gene prediction which were then used to train SNAP and AUGUSTUS. In addition, the following reference datasets were used: CD-HIT-clustered RNA-seq transcripts for mangrove HSC; all chelicerate proteins from RefSeq that were filtered for uncharacterized, partial and fragmented proteins; combined set of 36,493 proteins for *Bombyx mori*, *Drosophila melanogaster* and *Apis mellifera* obtained from FlyBase (https://flybase.org/), and ~112,000 proteins from vertebrates and invertebrates from SWISSPROT. These transcript and protein datasets were aligned to the genome and used to generate hint files using the MAKER pipeline. The hint files were used as input to aid in the gene prediction process and to calculate the annotation edit distance (AED) score. Predicted protein sequences that had no similarity to any protein in the NCBI-NR database (BLASTP; E-value: $10^{-7}$) or had similarity with "low-complexity" proteins were removed from the final set. InterProScan v5.28-67.0[55] was used at default settings to identify domains and GO terms associated with the predicted mangrove HSC proteins.

**Neutral mutation rate.** Protein datasets from the following representative chelicerate species with whole-genome sequences were downloaded from various databases (see Supplementary Table 8): Atlantic horseshoe crab, common house spider, velvet spider, Brazilian white-knee tarantula, bark scorpion, black-legged tick, and two-spotted spider mite (scientific names given in Supplementary Table 8). Proteome dataset for the European centipede (Supplementary Table 8), was used as an outgroup for chelicerates. InParanoid version 4.1[56] was used to identify orthologues with mangrove horseshoe crab proteins as reference and identified 162 strict one-to-one orthologues for the 9 taxa.

Multiple alignments of the protein datasets were generated using Muscle version 3.7 (https://www.ebi.ac.uk/Tools/msa/muscle/). A concatenated protein alignment was prepared and the best-suited substitution model was deduced using ModelFinder[57]. IQ-TREE version 1.6.10[58] was used to generate a maximum likelihood (ML) tree. For the ML analyses, we used the ModelFinder + tree reconstruction + non-parametric bootstrap option with 100 bootstrap replicates for node support. The best-fit substitution model according to Bayesian information criterion, as deduced by ModelFinder (LG+F+I+G4), was used for phylogenetic analysis.

A neutral tree based on an alignment of four-fold degenerate (4D) sites was generated. We used the topology obtained from our phylogenomic analyses as well as another based on the latest published topology[59] as an input for RAxML-based optimization of the branch lengths for the 4D alignment. Codon alignments of the coding sequences were generated from protein alignments using PAL2NAL[60]. A concatenated coding sequence alignment was generated and an alignment of 4D sites was extracted using the RPHAST package[61]. We used the "-f e" option in RAxML-8.1.3[62] to generate neutral trees for the 4D alignment. The neutral mutation rate was calculated using the branch lengths of the neutral tree and a divergence time of 135 million years between the mangrove and Atlantic HSC[39]. We obtained nearly identical neutral mutation rate estimates using the two topologies ($1.37 \times 10^{-9}$ substitutions/site/year and $1.38 \times 10^{-9}$ substitutions/site/year, respectively).

**Pairwise sequentially Markovian coalescent (PSMC) analysis.** We performed a quality check of the raw reads in FastQC (https://www.bioinformatics.babraham.ac.uk/projects/fastqc/), and mapped the reads to the mitogenome of the mangrove horseshoe crab (assembled in-house) using BWA-MEM 0.7.7-r441[63] to remove mitogenomic information from our analyses. We then filtered unmapped reads

through samtools 0.1.19[64,65] for further use. However, lack of information regarding sex chromosomes in HSCs prevented us from filtering our reads for sex chromosomal segments. Hence, our historical demographic reconstructions might include biases arising from sex-specific evolutionary trajectories.

We mapped the filtered reads to the assembled genome using BWA-MEM. For this purpose, we only retained reads with a high mapping score (greater than 20) and sorted the bam files in PICARDTOOLS 1.95 (http://broadinstitute.github.io/picard). Variable sites were identified using samtools mpileup and bcftools. For SNP calling we used the following parameters: -C 50, -d 10 and -D 250. The PSMC analysis consisted of the following parameters: -t 15 -r 5 -p 4+25*2+4+6, and comprised 30 iterations for parameter optimization and 100 bootstraps to obtain a measure of uncertainty around parameter estimates. Effective population size was calculated using a mutation rate of $1.37 \times 10^{-9}$ substitutions per site per year and generation time of 14 years.

**Hox gene clusters**. In addition to MAKER annotation, the Hox gene clusters were manually annotated to obtain the complete gene set. An alignment of Hox proteins from mangrove HSC, Atlantic HSC, house spider, bark scorpion, *Daphnia* or *Drosophila* was generated using ClustalW (http://www.clustal.org/clustal2/) as implemented in BioEdit sequence alignment editor[66]. A Maximum Likelihood was generated using IQ-TREE version 1.5.6[58] with a JTT+F+I+G4 substitution model as deduced by ModelFinder[57] and 1000 ultrafast bootstrap replicates.

**Comparison of mangrove HSC and tick genomes**. The protein dataset for the black-legged tick (*Ixodes scapularis*) (accession number GCA_002892825.2) was obtained from the NCBI Genomes FTP site. The longest representative isoform for each gene was extracted using an in-house script. In order to identify orthologues between mangrove HSC and tick, we ran InParanoid version 4.1[56] at default settings. The InParanoid table output file was used as an input for i-ADHoRe version 3.0.01[23] in order to remove tandem duplicates. We selected only those families containing one member in tick and up to eight members in the mangrove HSC.

**Paralogous gene segments in the mangrove HSC genome**. In order to identify paralogues in the mangrove HSC genome, proteins filtered for tandem duplicates were used for BLASTP at an E-value cut-off of 1e−8. The BLAST output was clustered using the program MCL (Markov Cluster Algorithm) version 14-137[67] with an 'inflation' parameter of six. The final clustered output file was filtered to retain only groups containing eight HSC members or less. These groups were processed using i-ADHoRe which also removes tandem duplicates. The syntenic segments were visualized using Circos version 0.69 (https://github.com/vigsterkr/circos).

**Estimation of transversions at four-fold degenerate sites**. To identify mangrove HSC paralogues, we performed self-BLASTP of the proteins (filtered for tandem duplicates) using an E-value threshold of 1e−8. The generated output file was used for clustering using MCL version 14-137[67] at an inflation parameter of six. The clusters were filtered to retain families with two to eight members. Amino acid alignments were generated for all paralogous pairs using ClustalW 2.1 (http://www.clustal.org/clustal2/) and coding sequences were aligned using PAL2NAL version 14[60] based on the protein alignments. Four-fold degenerate (4D) sites were batch extracted from the coding sequence alignments using the RPHAST package (R version 3.5.1)[61]. Only sequence pairs containing ≥30 4D sites (10,390 gene pairs) were retained for further analysis. The number of transversions (Tv) per sequence were calculated for the 4D alignments using another in-house Perl script. The 4DTv rate was calculated as the number of transversions divided by the number of 4D sites. The 4DTv rates were corrected for possible multiple substitutions (transversions) using the following formula as per a previous study[68]:

$$4\mathrm{DTv}_{corrected} = -1/2 \ln(1 - 2 \times 4\mathrm{DTv}_{uncorrected})$$

By using Mclust program (https://mclust-org.github.io/mclust/), several univariate normal mixture models with varying numbers of population components were fitted to the distribution of corrected 4DTv rates and the best-fitted model was identified on the basis of Bayesian Information Criterion (BIC).

**Identification of tandem gene clusters in the assembly**. An in-house program was developed to identify clusters of tandem duplicate genes in the mangrove HSC genome. First, we identified all genes with the same gene name (gene description) present in tandem that are not separated by more than five unrelated genes. To ensure that they are indeed tandem duplicates, we require that tandem copies should be at least 70% identical to each other. Finally, we confirmed the identities of the tandem copies by BLASTP against the reference protein dataset that was used for genome annotation. Tandem gene copies typically match the same protein in the reference dataset. Clusters of three or more genes were considered as 'tandem gene clusters'. Selected gene clusters (*carcinolectin-5b*, *carcinolectin-3/CEBP-PI* and *FucTC*) were manually curated which helped in identifying additional genes than those predicted by the MAKER pipeline. The GO enrichment analysis was performed using Fisher's Exact Test from Blast2GO package version 5.2.5[69]. FDR was calculated as an adjustment for multiple comparisons. All proteins corresponding to genes in the tandem gene clusters were used as 'test dataset' whereas

rest of the predicted proteins were used as 'reference dataset'. The proteins with at least one GO term predicted were used for the analysis.

**Reporting summary**. Further information on research design is available in the Nature Research Reporting Summary linked to this article.

## Data availability
The whole-genome sequence of the mangrove horseshoe crab has been deposited in the DDBJ/EMBL/GenBank database under the accession number VWRL01. RNA-seq reads for the four tissues of the mangrove horseshoe crab have been deposited in the NCBI Sequence Read Archive under accession number SRP139459.

## Code availability
All custom codes used in this study are available upon request.

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

## Acknowledgements

We acknowledge the National Supercomputing Centre of Singapore for providing computational resources. This research is supported by the National Research Foundation, Prime Minister's Office, Singapore under its Marine Science Research and Development Programme (Award No. MSRDP-P19) and MOE Tier II Grant (R-154-000-A59-112). B.C. acknowledges funding from the South East Asian Biodiversity Genomics (SEABIG) Grant (number WBS R-154-000-648-646 and WBS R-154-000-648-733).

## Author contributions

B.V. conceived, designed, and coordinated the project. L.-M.Y. provided the mangrove horseshoe crab specimen; B.V. and B.-H.T. prepared the DNA and RNA samples and libraries; P.S., V.R., A.P., and B.V. performed genome analyses; K.M.G., B.C., and F.E.R. performed the PSMC analysis. B.V., V.R., and P.S. wrote the paper with inputs from other authors.

## Competing interests

The authors declare no competing interests.
