## [Peer Review File · Nature Communications]

Reviewers' comments:

Reviewer #1 (Remarks to the Author):

Gene duplication is thought to generate genetic material to facilitate organismal innovation and diversification. It follows that whole genome duplication (WGD) is often thought to have contributed to the evolutionary success of animals like vertebrates. However, relatively few WGD events have been found so far in animals and therefore it is debated if these events necessarily beget diversification or indeed are needed for diversification. Therefore, it is necessary to identify WGD events that have occurred during animal evolution, especially independent events from vertebrate WGDs, and to assess their potential contribution to diversification.

Horse-shoe crabs (HSCs) are a very interesting case. There are only four extant species of these animals, but it was previously reported that they had been subject to two ancestral WGDs (Kenny et al.,). Furthermore, it appeared that these two WGD events in HSCs were independent of a single WGD in the ancestor of other chelicerates – the arachnids, spiders and scorpions (Schwager et al.,). However previously assemblies of the HSC genomes have been very fragmented and assessment of the outcomes of WGD requires analysis of genes synteny. For many species this has only recently become possible using combinations of long read technologies and Hi-C and therefore this work on the improved assembly of the genome of the mangrove HSC (to basically the chromosomal level) is timely and provides important new insights into the biology of these animals and the outcomes of WGD more broadly. It is therefore an important study that contributes greatly to our understanding of animal evolution.

The improved assembly provided by the authors allows them to provide a more detailed understanding of the outcome of reiterative WGD in HSCs and in fact they find solid evidence for the occurrence of not two but potentially three WGDs in these animals. The main evidence for this is their finding of five clusters of Hox genes as well as duplicated clusters of Irx genes and microRNAs. In addition, comparisons to the tick genome allowed the authors to identify many other clusters of genes that have been retained as duplicates since these WGD events. They also provide further evidence that these WGD events are independent from WGD in spiders and scorpions. GO analysis of duplicated genes in HSC suggests pervasive retention of genes involved in immunity and thus new insights into the biology of these animals. Furthermore, population genomics analyses allowed the authors to study population history of these animals and pinpoint a reduction in the population size consistent with the start of the most recent ice age.

The paper is well written and makes an important contribution to our understanding of WGDs and animal evolution. It offers a key counterpoint to the paradigmatic view of WGD leading to diversification as is widely held to be the case in vertebrates. This paper will also serve as a platform to further explore the consequences of WGD in these animals, other chelicerates and other metazoans.

I have no major concerns with the paper, but as the authors themselves are very careful throughout the manuscript to weigh the evidence for two versus three WGDs in HSCs, and since this the evidence provided is not completely conclusive, it might better to adjust the title accordingly.

One other minor point is that figure 3 suggests the Hox clusters are known to be intact in spiders and scorpions - however one of the clusters of the spider has only been assembled into two fragments so far while the Hox clusters of the scorpion are still quite fragmented in the most recent assemblies.

Reviewer #2 (Remarks to the Author):

General comments

The manuscript presents a high-quality genome assembly for the mangrove horse-shoe crab, one of four extant horse-shoe crab species. The authors argue for three whole genome duplications (WGD), meaning that the genome is principally octaploid. However, three main reasons make me hesitate to draw such a conclusion:

1) No gene families are shown with more than five members, except for the mir-71 micro-RNA which has seven. This is despite the fact that the three WGD events are deduced to have happened 500-135 Mya. By comparison, teleosts that had their three WGD events in the range 550-300 Mya can have up to eight copies, i.e., a fully 'octaploid' (but diploidized) gene set. Salmonids and carps independently experienced a fourth WGD and can have several additional copies beyond eight.

2) Many tandem duplicates, or gene clusters, are described (Fig.8). If this happens with such high frequency, perhaps independent duplications followed by translocations may explain some or most of the copies interpreted to be the result of WGD events.

3) The tick chromosomes in Fig. 5 consistently show high similarity to a single HSC chromosome each and then dramatically lower similarity to other HSC chromosomes. Why would only a single HSC chromosome remain reasonably intact after the three proposed WGD events? For comparison, consider the analysis of a lamprey genome by Smith et al. 2018 (PMID 29358652) where a similar chart in Fig. 4 conspicuously showed that many lamprey super-scaffolds displayed similarity to 2-4 chicken chromosomes, cautiously interpreted as one WGD, but most likely two WGD events. For the vertebrates, these two events took place >500 Mya and the pattern is still quite obvious. Why does the HSC genome show only a single chromosome with high similarity to tick in each and every case although its proposed WGD events are more recent?

The manuscript is overall well written and easy to follow, but a few important pieces of information are missing and I am not convinced by the overall conclusions, as detailed below.

Specific points

Page 3 and Suppl. Fig. 1: The interpretation of the k-mer data is not described in detail and no reference is given. Please explain how the K-mer copy number distribution can provide the genome size and the heterozygosity (but apparently not the genome doublings).

Page 5 and Suppl. Fig. 2: The calculated mutation rate must relate to some kind of time estimate for the species divergencies. From the fossil record? Please clarify.

Page 5: The two basal vertebrate WGD events were shown much more clearly by Nakatani et al. 2007 (PMID 17652425) and Putnam et al. 2008 (18563158) than by Dehal & Boore who showed a number of theoretical figures and then only studied the human genome and who did not even include human chromosome 7 with the HoxA cluster in their Fig. 7.

Page 6: The discussion about the Irx gene clusters should mention the possibility that the Sowah-like gene in IRX-E on Chr4 may have been translocated from either IRX-C or IRX-D.

Page 7 and Fig. 3B: Likewise, the chromosomes maps may involve a fission resulting in the two genes on Chr15 and the two genes in the second block on Chr1.

Page 7 and Fig. 5A: It is very clear that each tick contig has a 'favorite' HSC chromosome. If octaploidization has taken place in HSC, why has always one chromosome been maintained much better than the other seven, as I commented in my point 3 above?

Page 7 and Fig. 5B: It is not easy to see if single genes in Ixodes corresponds to up to six genes HSC. Rather, the Ixodes genes seem to have been dispersed by chromosome fissions onto the six

chromosomes in HSC.

Page 8 and Fig. 6: Again, why is this analysis interpreted to mean up to 8 copies of each block rather than a lot of fissions?

Page 8 and Fig. 7: How many gene pairs were included in this analysis? My immediate impression when seeing the black plot was that there was one extensive duplication phase (WGD) preceded by extended duplications over a long period of time. The data described for tandem duplications would seem consistent with the possibility of extensive independent duplications rather than additional WGD events.

Page 10, bottom line: the authors write that there is 'strong evidence for three rounds of WGD'. I am afraid I am not as convinced as the authors, as described above.

Response to reviewers' comments

Reviewer #1 (Remarks to the Author):

Gene duplication is thought to generate genetic material to facilitate organismal innovation and diversification. It follows that whole genome duplication (WGD) is often thought to have contributed to the evolutionary success of animals like vertebrates. However, relatively few WGD events have been found so far in animals and therefore it is debated if these events necessarily beget diversification or indeed are needed for diversification. Therefore, it is necessary to identify WGD events that have occurred during animal evolution, especially independent events from vertebrate WGDs, and to assess their potential contribution to diversification.

Horse-shoe crabs (HSCs) are a very interesting case. There are only four extant species of these animals, but it was previously reported that they had been subject to two ancestral WGDs (Kenny et al.). Furthermore, it appeared that these two WGD events in HSCs were independent of a single WGD in the ancestor of other chelicerates – the arachnids, spiders and scorpions (Schwager et al.). However previously assemblies of the HSC genomes have been very fragmented and assessment of the outcomes of WGD requires analysis of genes synteny. For many species this has only recently become possible using combinations of long read technologies and Hi-C and therefore this work on the improved assembly of the genome of the mangrove HSC (to basically the chromosomal level) is timely and provides important new insights into the biology of these animals and the outcomes of WGD more broadly. It is therefore an important study that contributes greatly to our understanding of animal evolution.

Response: We thank the reviewer for the encouraging comments.

The improved assembly provided by the authors allows them to provide a more detailed understanding of the outcome of reiterative WGD in HSCs and in fact they find solid evidence for the occurrence of not two but potentially three WGDs in these animals. The main evidence for this is their finding of five clusters of Hox genes as well as duplicated clusters of Irx genes and microRNAs. In addition, comparisons to the tick genome allowed the authors to identify many other clusters of genes that have been retained as duplicates since these WGD events. They also provide further evidence that these WGD events are independent from WGD in spiders and scorpions. GO analysis of duplicated genes in HSC suggests pervasive retention of genes involved in immunity and thus new insights into the biology of these animals. Furthermore, population genomics analyses allowed the authors to study population history of these animals and pinpoint a reduction in the population size consistent with the start of the most recent ice age.

The paper is well written and makes an important contribution to our understanding of WGDs and animal evolution. It offers a key counterpoint to the paradigmatic view of WGD leading to

diversification as is widely held to be the case in vertebrates. This paper will also serve as a platform to further explore the consequences of WGD in these animals, other chelicerates and other metazoans.

Response: We thank the reviewer for the positive comments.

I have no major concerns with the paper, but as the authors themselves are very careful throughout the manuscript to weigh the evidence for two versus three WGDs in HSCs, and since this the evidence provided is not completely conclusive, it might better to adjust the title accordingly.

Response: We have now provided further evidence that strongly favours three WGDs (10 new paralogous gene loci with 6 to 8 copies. See new Fig. 5 and Supplementary Fig. 4 to 13). Nevertheless, we have modified the title to “Chromosome-level assembly of the horseshoe crab genome provides new insights into its genome evolution”.

One other minor point is that figure 3 suggests the Hox clusters are known to be intact in spiders and scorpions - however one of the clusters of the spider has only been assembled into two fragments so far while the Hox clusters of the scorpion are still quite fragmented in the most recent assemblies.

Response: We have revised Figure 3a to reflect the fragmented nature of the spider and scorpion Hox clusters in their genome assemblies.

Reviewer #2 (Remarks to the Author):

General comments

The manuscript presents a high-quality genome assembly for the mangrove horse-shoe crab, one of four extant horse-shoe crab species. The authors argue for three whole genome duplications (WGD), meaning that the genome is principally octaploid. However, three main reasons make me hesitate to draw such a conclusion:

1) No gene families are shown with more than five members, except for the mir-71 micro-RNA which has seven. This is despite the fact that the three WGD events are deduced to have happened 500-135 Mya. By comparison, teleosts that had their three WGD events in the range 550-300 Mya can have up to eight copies, i.e., a fully 'octaploid' (but diploidized) gene set. Salmonids and carps independently experienced a fourth WGD and can have several additional copies beyond eight.

Response: Thanks for highlighting the lack of examples of paralogous genes with more than five members which would be expected if the genome had experienced three WGD events. We have now further analyzed the genome and present 10 examples of paralogous gene loci with seven and six copies including one with potentially eight copies (see new Fig. 5 and Supplementary Figures 4 to 13). As mentioned in the revised Results section, the presence of six to eight copies of these paralogous loci comprising multiple unrelated genes (i.e. genes that are not the result of tandem duplications like the Hox clusters) on mostly different chromosomes is more consistent with three WGD events. An examination of these loci shows that only one or two genes in each loci is present in six or seven copies while the other genes are present in only two to five copies suggesting that an extensive secondary loss of duplicated genes occurred after the WGD events. This also explains why we do not find many paralogous gene loci with more than five copies.

We agree that the proposed three WGD events in the horseshoe crab parallels the 3R in teleosts (i.e. 2R in the ancestor of jawed vertebrates plus a third R in the teleost ancestor). However, we would like to point out that in teleosts not many paralogous gene loci are present in eight copies and Hox cluster is one such exception that is present in eight copies. In fact, only two lineages of teleosts, the eels and bonytongues (see Bian et al. 2016, The Asian arowana genome provides new insights into the evolution of an early lineage of teleosts. *Sci. Rep.* 6: 24501) possess eight Hox clusters whereas the rest possess only seven clusters (some lineages like salmonids, which have experienced 4R, possess more than eight clusters). It is also possible that the three WGD events in horseshoe crabs might have occurred much earlier than the 3rd R in teleosts which is estimated to have occurred around 300 - 350 million years ago. In addition, it is likely that the duplicated genes are under different evolutionary selection in the teleosts and horseshoe crab lineages, which may explain the presence of ~30,000 extant species of teleosts compared to only four horseshoe crab species.

2) Many tandem duplicates, or gene clusters, are described (Fig.8). If this happens with such high frequency, perhaps independent duplications followed by translocations may explain some or most of the copies interpreted to be the result of WGD events.

Response: We agree that this is one possibility. However, if gene duplication had occurred independently and then translocated to a different chromosome, such paralogues would be distributed randomly in the genome and would not have ended up as syntenic blocks of genes on multiple chromosomes. The new examples of paralogous gene loci that we have now presented with six to eight copies of clusters of unrelated genes present on different chromosomes shows that these copies are more likely to be due to WGD. They are unlikely to be the result of tandem duplication and translocations. As explained below, our Circos plot of paralogous gene loci in mangrove horseshoe crab further argues in favour of WGD events rather than tandem duplications and translocation.

3) The tick chromosomes in Fig. 5 consistently show high similarity to a single HSC chromosome each and then dramatically lower similarity to other HSC chromosomes. Why would only a single HSC chromosome remain reasonably intact after the three proposed WGD events? For comparison, consider the analysis of a lamprey genome by Smith et al. 2018 (PMID 29358652) where a similar chart in Fig. 4 conspicuously showed that many lamprey super-scaffolds displayed similarity to 2-4 chicken chromosomes, cautiously interpreted as one WGD, but most likely two WGD events. For the vertebrates, these two events took place >500 Mya and the pattern is still quite obvious. Why does the HSC genome show only a single chromosome with high similarity to tick in each and every case although its proposed WGD events are more recent?

Response: We thank the reviewer for this comment. We realize that the way we have presented our data in Fig. 5a gives the false impression of the “tick chromosomes consistently showing high similarity to a single HSC chromosome each”. First, we would like to clarify that the tick genome assembly is at contig level (with a contig N50 of 835 kb) and not at chromosome level like the HSC or lamprey genome assemblies. Each tick contig contains far fewer genes than the HSC chromosomes. Second, of the 24,054 genes in the tick genome, only 5,286 possess one to eight orthologues (the expected number of products of three WGD events) in the HSC genome. In fact, a majority of these genes (3,825 genes) have only 1-to-1 orthologues while the rest possess two to eight orthologues (972, 1-to-2; 319, 1-to-3; 115, 1-to-4; 40, 1-to-5; 10, 1-to-6; 5, 1-to-7; and 0, 1-to-8) in the HSC genome. As can be seen, only a small number of tick genes have 2-to-multiple orthologues in the horseshoe crab genome and these genes are scattered on several tick contigs. Thus, there is very limited syntenic information between the tick and HSC genomes. Third, the black horizontal lines in original Fig. 5A are not tick chromosomes or contigs but are in fact the boundaries of groups of contigs (note that each contig is represented by green horizontal lines, which unfortunately are barely visible due to the faint colour used). Fourth, the cluster of dots along the diagonal does not mean that the tick orthologues on a single contig map mainly to a single HSC chromosome. If one observes carefully, one can see that the first HSC chromosome contains orthologues from many tick contigs (182 to be precise, represented by green horizontal lines from the bottom to the first black horizontal line). The dotplot with clusters of genes spread across the diagonal gives the impression that each tick contig contains orthologues for mainly one HSC chromosome. This is because of the way we had clustered (as mentioned in the original Methods section and legend for original Figure 5A) the tick contigs into 16 groups

(black horizontal lines) according to the highest number of orthologues for each the 16 HSC chromosomes. This gives an erroneous impression. If we had presented the contigs along the Y-axis according to their size (as shown in the right image below), then tick genes on a contig do not show “preference” to any single HSC chromosome but instead are scattered on all chromosomes.

We realize that this data (with limited syntenic information between tick and HSC) is not informative and gives a wrong impression about orthologous relationships between tick and HSC genomes. Therefore, we have now removed the dot plot (old Fig. 5A) from the manuscript and instead present the actual data of orthologous genes between the tick and HSC genomes (page 7, 2nd para – section “Comparison with the tick genome”). In addition, we have added two more examples of syntenic gene distribution between the tick contigs and HSC chromosomes to the old Fig. 5B (which is now new Fig 6a-c).

The manuscript is overall well written and easy to follow, but a few important pieces of information are missing and I am not convinced by the overall conclusions, as detailed below.

Specific points

(i) Page 3 and Suppl. Fig. 1: The interpretation of the k-mer data is not described in detail and no reference is given. Please explain how the K-mer copy number distribution can provide the genome size and the heterozygosity (but apparently not the genome doublings).

Response: The first peak of the k-mer plot (KCN=46) represents the heterozygous single copy k-mers while the second peak (KCN=92) represents the homozygous single copy k-mers in the genome. We have now added this description in the manuscript (page 16, 1st para – section “Genome size and heterozygosity level” within Methods) and in the legend for Supplementary Fig. 1. There is a standard formula to calculate the genome size using k-mer

statistics developed by Lander and Waterman (Lander and Waterman, 1988 Genomic mapping by fingerprinting random clones: a mathematical analysis. *Genomics*, 2(3): 231-239). We have now cited this reference. As mentioned in the manuscript, the heterozygosity level was estimated using the tool “GenomeScope” (Vurture et al., 2017 *Bioinformatics* 33: 2202-2204) which is an open-source tool to rapidly estimate the heterozygosity level. GenomeScope fits a mixture model of four evenly spaced binomial distributions to the k-mer count distribution to measure the relative abundances of heterozygous and homozygous sequences in the genome (Vurture et al. 2017). The k-mer plot will not be able to capture the WGD events as these events occurred more than 150 years ago and the duplicate copies of genes have diverged much more than the two alleles in the genome.

(ii) Page 5 and Suppl. Fig. 2: The calculated mutation rate must relate to some kind of time estimate for the species divergencies. From the fossil record? Please clarify.

Response: We thank the reviewer for pointing this out. The mutation rate was indeed calculated using the divergence time between mangrove and Atlantic horseshoe crab, i.e. 135 million years, as deduced by Obst et al. (Molecular phylogeny of extant horseshoe crabs indicates Paleogene diversification of Asian species. *Mol. Phylogen. Evol.* 2012. 62: 21-26). The divergence times estimated in this study were based on 18S, 28S and COI sequences and calibration was based on fossil records of *Mesolimulus walchi*, a species often assumed to represent the stem group of all horseshoe crabs. We have now mentioned the divergence time and the relevant reference in the Methods section (page 17, 3rd para of “Neutral mutation rate” section).

(iii) Page 5: The two basal vertebrate WGD events were shown much more clearly by Nakatani et al. 2007 (PMID 17652425) and Putnam et al. 2008 (18563158) than by Dehal & Boore who showed a number of theoretical figures and then only studied the human genome and who did not even include human chromosome 7 with the HoxA cluster in their Fig. 7.

Response: We thank the reviewer for this suggestion. We have replaced Dehal & Boore (2005) with the two references mentioned by the reviewer.

(iv) Page 6: The discussion about the *Irx* gene clusters should mention the possibility that the Sowah-like gene in IRX-E on Chr4 may have been translocated from either IRX-C or IRX-D.

Response: We agree with the reviewer. It is indeed possible that the Sowah-like gene in IRX-E on Chr4 may have been translocated from IRX-C or IRX-D, in which case there would be only four *Irx* loci. However, since we have now included several new and better examples of paralogous loci with six to seven copies, we have removed the *Irx* gene data from the manuscript.

(v) Page 7 and Fig. 3B: Likewise, the chromosomes maps may involve a fission resulting in the two genes on Chr15 and the two genes in the second block on Chr1.

Response: Yes, it is possible that the two microRNA genes (*mir-71* + *mir-2*) on Chr15 were initially part of the second block of *mir-2* genes on Chr1, which translocated following

fission of the chromosome. We have now mentioned this possibility in the text (page 6, 3rd para) and have accordingly concluded that this would result in seven loci.

(vi) Page 7 and Fig. 5A: It is very clear that each tick contig has a 'favorite' HSC chromosome. If octaploidization has taken place in HSC, why has always one chromosome been maintained much better than the other seven, as I commented in my point 3 above?

Response: Thanks for pointing out this figure which inadvertently gave a misleading interpretation. Please see the detailed clarification we have provided about this figure in our response to your point #3 above.

(vii) Page 7 and Fig. 5B: It is not easy to see if single genes in Ixodes corresponds to up to six genes HSC. Rather, the Ixodes genes seem to have been dispersed by chromosome fissions onto the six chromosomes in HSC.

Response: We beg to differ with the reviewer's interpretation of this figure and would like to highlight that in Figure 5B, each gene in Ixodes is not mapping to six genes in mangrove HSC. Rather, genes present on a single tick contig are mapping to their orthologues present on six mangrove HSC chromosomes. In case of a chromosomal fission scenario, we would expect to see a pattern whereby genes present on one portion of the tick contig map to one mangrove HSC chromosome whereas genes present on another region of the tick contig would map to a different mangrove HSC chromosome. We do not see such a pattern in our case. The interdigitated distribution of HSC orthologues on multiple chromosomes is suggestive of WGD followed by random loss of duplicated genes from each chromosome. We have now provided two more such examples (see new Fig. 6a-c).

(viii) Page 8 and Fig. 6: Again, why is this analysis interpreted to mean up to 8 copies of each block rather than a lot of fissions?

Response: We again beg to differ with the reviewer's interpretation, which is incorrect. First, we would like to clarify that the bands represent "anchor points" which are pairs of statistically significant collinear paralogous genes residing within paralogous regions on pairs of HSC chromosomes. In other words, these are pairs of paralogous genes shared between HSC chromosomes. The anchor points were identified by the program i-ADHORE based on conservation of gene content and gene order across the entire chromosomes. The thickness of the bands is proportional to the number of paralogous genes shared between a pair of chromosomes. Please note that the position of the bands on the chromosomes do not indicate the location of genes on the chromosomes. We have now made this clear in the figure legend. The paralogous genes could be spread over a long region. Thus, the bands do not indicate chromosomal fissions as interpreted by the reviewer.

We agree that it is not appropriate to specify "up to 8 copies" as not the same set of syntenic blocks of genes are shared among the 8 chromosomes. The important feature is the distribution pattern of paralogous syntenic blocks across the genome. In case the duplicate genes were the result of independent tandem duplications followed by translocation, the paralogues would have been randomly distributed across the genome instead of blocks of

syntenic genes between chromosomes as shown in the figure. In the alternative probable scenario of duplication of only one or two chromosomes instead of WGD, there would be a large number of paralogous syntenic genes shared between just one or two pairs of chromosomes, which is also not the case here. The distribution of paralogous syntenic blocks of genes between several chromosomes seen in the figure is more consistent with WGD rather than tandem gene duplication or chromosomal duplication. We have now rewritten this section to highlight that the pattern of distribution of paralogous syntenic blocks of genes is consistent with WGD (page 8, 1st para – section “Paralogous segments in the mangrove HSC genome”).

(ix) Page 8 and Fig. 7: How many gene pairs were included in this analysis? My immediate impression when seeing the black plot was that there was one extensive duplication phase (WGD) preceded by extended duplications over a long period of time. The data described for tandem duplications would seem consistent with the possibility of extensive independent duplications rather than additional WGD events.

Response: We used in total 10,390 gene pairs to generate the 4DTv plot (Figure 7). This is after removal of protein sequences containing 30 or fewer 4D sites from a larger set (>18,000 gene pairs). We have now given these numbers in the Methods section (page 19, section “Estimation of transversions at four-fold degenerate sites (4DTv)”). The black trendline plot shows one major peak, a medium size peak and a possible minor peak or a shoulder peak. In order to determine the exact number of peaks, we used the Mclust program to fit a range of univariate models and let the Bayesian Information Criterion (BIC) select the best-fit model that accommodates most of the data (BIC is routinely used for generating best-fit models of trendlines e.g., Nossa et al., 2014; Atlantic HSC genome paper, Fig. 10). The best-fit model comprised four distinct population components of which the first peak corresponds to recent small-scale duplications (not shown in old Fig. 7, now Figure 8) and the remaining three peaks represent three populations of paralogous genes (green, blue and red peaks in Figure 7). This suggests three rounds of WGD.

(x) Page 10, bottom line: the authors write that there is 'strong evidence for three rounds of WGD'. I am afraid I am not as convinced as the authors, as described above.

Response: We understand the reviewer's concern and hope that with the clarifications and additional data provided, the reviewer will be convinced that the data presented is more consistent with three WGD events rather than other possibilities (independent tandem duplications or chromosomal duplication). Nevertheless, we have toned down the above-mentioned statement on page 11 (Discussion section) to “..argue more in favour of three rounds of WGD in the HSC lineage than independent tandem duplications followed by translocation or large-scale segmental duplications”. Likewise, we have also toned down our statement in the Abstract (revised “provides evidence for three rounds of WGD” to “provides evidence that suggests three rounds of WGD”). In addition, we have revised the title of the manuscript as “Chromosome-level assembly of the horseshoe crab genome provides new insights into its genome evolution”.

REVIEWERS' COMMENTS:

Reviewer #1 (Remarks to the Author):

This is an interesting manuscript that provides important new insights into the outcomes of whole genome duplications in animals. My previous review of the manuscript highlighted a few minor concerns that the authors have now addressed.

Reviewer #2 (Remarks to the Author):

What a difference! The responses and clarifications by the authors have improved the manuscript dramatically. It is now much clearer and argues quite convincingly for three genome doublings in the horseshoe crab lineage. The statistics for multimember gene families now included provides valuable information. It is actually a rather small fraction of the tick genome that allows comparison with the HSC: 5286 genes out of 24,000. Only 56 of these have more than 4 co-orthologs in HSC.

However, two matters remain, namely concerning the new Fig. 6 and and Fig. 7, or rather the way they are described and interpreted. This should be possible to rectify by adjusting the text.

Regarding the authors' response to my question about Page 7 and Fig. 5B:

As the HSC and tick lineages diverged such a long time ago, the situation shown in the figure (now Fig. 6a) could have arisen by multiple rearrangements within the tick chromosome (and some in the HSC chromosomes), thereby reshuffling gene order. This is quite frequently observed among vertebrates lineages. The ancestral genome doublings can nevertheless be deduced in vertebrates by comparing many different species, representing deep lineages. This abundance of lineage representation is not available for the present comparison. The authors write in conjunction with the description of Fig. 6: "This pattern of interdigitated distribution of mangrove HSC orthologues on different chromosomes suggests that these syntenic blocks of genes in mangrove HSC are likely to be the result of WGD rather than segmental or tandem duplication events." I would suggest the authors tune down the argument by replacing "are likely to" to "may".

Regarding my question about page 8 and Fig. 6 (now Fig. 7), I had written: "Again, why is this analysis interpreted to mean up to 8 copies of each block rather than a lot of fissions?" I should have written: Why does this indicate 3 WGD events rather than only 1?

The authors' description of the figure now reads like this: "each mangrove HSC chromosome contained paralogs mapping to multiple other chromosomes. This pattern is more consistent with WGDs rather than independent chromosomal or tandem gene duplication events and supports the hypothesis that the HSC lineage has experienced three rounds of WGD."

However, I still cannot see how this analysis can say whether there were 1, 2 or 3 WGD events. Each region in the HSC genome is shown to one (1) similar region elsewhere in the genome. For those (small and few) blocks that have more than one duplicate, how did the analysis choose to display only one? And which one? I assume it is the most similar one (= the most recent one) that is caught with this analysis. Maybe the blocks from earlier WGD events are so small and few that they fall below the threshold of the analysis? As far as I can tell, the analysis displayed in Fig. 7 shows that the entire HSC genome has been duplicated, but it is impossible to tell from this analysis whether it happened more than once. Even the phrase "each mangrove HSC chromosome contained paralogs mapping to multiple other chromosomes" I find misleading. I think it would be more fair to say: each mangrove HSC chromosomal region displays similarity to one other chromosomal region, presumably reflecting the most recent WGD event.

RESPONSE TO REVIEWERS' COMMENTS:

Reviewer #1 (Remarks to the Author):

This is an interesting manuscript that provides important new insights into the outcomes of whole genome duplications in animals. My previous review of the manuscript highlighted a few minor concerns that the authors have now addressed.

Response: We are glad that the reviewer is satisfied with the revisions.

Reviewer #2 (Remarks to the Author):

What a difference! The responses and clarifications by the authors have improved the manuscript dramatically. It is now much clearer and argues quite convincingly for three genome doublings in the horseshoe crab lineage. The statistics for multimember gene families now included provides valuable information. It is actually a rather small fraction of the tick genome that allows comparison with the HSC: 5286 genes out of 24,000. Only 56 of these have more than 4 co-orthologs in HSC.

However, two matters remain, namely concerning the new Fig. 6 and and Fig. 7, or rather the way they are described and interpreted. This should be possible to rectify by adjusting the text.

Response: We thank the reviewer for helping in substantially improving this manuscript. We have addressed the two remaining concerns of the reviewer in our responses below.

1. Regarding the authors' response to my question about Page 7 and Fig. 5B:

As the HSC and tick lineages diverged such a long time ago, the situation shown in the figure (now Fig. 6a) could have arisen by multiple rearrangements within the tick chromosome (and some in the HSC chromosomes), thereby reshuffling gene order. This is quite frequently observed among vertebrates lineages. The ancestral genome doublings can nevertheless be deduced in vertebrates by comparing many different species, representing deep lineages. This abundance of lineage representation is not available for the present comparison. The authors write in conjunction with the description of Fig. 6: "This pattern of interdigitated distribution of mangrove HSC orthologues on different chromosomes suggests that these syntenic blocks of genes in mangrove HSC are likely to be the result of WGD rather than segmental or tandem duplication events." I would suggest the authors tune down the argument by replacing "are likely to" to "may".

Response: Suggested change incorporated – “are likely to” has been replaced by “may” (page 7, last paragraph).

2. Regarding my question about page 8 and Fig. 6 (now Fig. 7), I had written: "Again, why is this analysis interpreted to mean up to 8 copies of each block rather than a lot of fissions?" I should have written: Why does this indicate 3 WGD events rather than only 1?

The authors' description of the figure now reads like this: "each mangrove HSC chromosome contained paralogs mapping to multiple other chromosomes. This pattern is more consistent with WGDs rather than independent chromosomal or tandem gene duplication events and supports the hypothesis that the HSC lineage has experienced three rounds of WGD." However, I still cannot see how this analysis can say whether there were 1, 2 or 3

WGD events. Each region in the HSC genome is shown to one (1) similar region elsewhere in the genome. For those (small and few) blocks that have more than one duplicate, how did the analysis choose to display only one? And which one? I assume it is the most similar one (= the most recent one) that is caught with this analysis. Maybe the blocks from earlier WGD events are so small and few that they fall below the threshold of the analysis? As far as I can tell, the analysis displayed in Fig. 7 shows that the entire HSC genome has been duplicated, but it is impossible to tell from this analysis whether it happened more than once. Even the phrase "each mangrove HSC chromosome contained paralogs mapping to multiple other chromosomes" I find misleading. I think it would be more fair to say: each mangrove HSC chromosomal region displays similarity to one other chromosomal region, presumably reflecting the most recent WGD event.

Response: The Circos plot (new Fig. 7) is based on all paralogous gene pairs present in the genome and includes recent as well as ancient paralogues. Please note that each band connecting two chromosomes does not represent a unique set of paralogous genes. For example, in Figure 7, chromosome 1 has a band connecting to chromosome 2 and another one connecting to chromosome 3. Some of chr1 genes that are part of the first band (connecting to chr2) can also be part of the second band (connecting to chr3). In order to further illustrate this, we present below Fig. 5b with connector lines: in this figure, chromosome 2 has paralogues on five other chromosomes (chr3, 9, 10, 12 and 13). If these paralogues were represented on a Circos plot, chromosome 2 would have five bands connecting to each of chromosomes 3 (NotchL and two copies of Frq1L), 9 (Gapr1L and NotchL), 10 (Gapr1L, NotchL, and DnmL), 12 (Gapr1L, NotchL, DnmL and two copies of Frq1L) and 13 (NotchL and Frq1L). Likewise, other chromosomes will have bands connecting to all other chromosomes. The bands we see in Fig. 7 are a composite of all paralogous genes between chromosomes. Thus, these bands show that each mangrove HSC chromosome contains genes that have paralogues on multiple other chromosomes. We agree that this does not suggest there were three WGD events, but it does suggest that there were WGD events rather than tandem duplications or chromosomal duplications. We have now made the statement mentioned by the reviewer (“..each mangrove HSC chromosome contained paralogs mapping to multiple other chromosomes.”) clearer by rephrasing it as “...each mangrove HSC chromosome contains genes whose paralogues map to multiple other chromosomes” (page 8, 1st paragraph).